# VARIATIONAL IMPLICIT DISTRIBUTION MATCHING

**Hiroshi Wakimori, Hiromasa Takei & Tikara Hosino**
Nihon Unisys, Ltd.
1-1-1 Toyosu Koto-ku, Tokyo 135-8560 Japan

{hiroshi.wakimori,hiromasa.takei,chikara.hoshino}@unisys.co.jp

## ABSTRACT

Modeling with implicit distributions such as GAN is effective technique in the probabilistic generative models because of its flexibility and few assumptions on the target distribution. For associating two distributions, ALI and other variants were proposed with joint distribution matching. However, they have disadvantages in the balance of the mode missing behavior and sample reconstruction quality. In this paper, we propose variational implicit distribution matching which is natural probabilistic extension of joint matching to its marginal and conditional distributions. We experimentally show that our proposal achieves the balance of the reconstruction and random generation quality and is competitive to the state of the art.

## 1 INTRODUCTION

Associating the two distributions is a fundamental problem in the deep generative models. For example, GAN (Goodfellow et al., 2014) and VAE (Kingma & Welling, 2013) associate the sample $x$ distribution and the hidden variable $z$ distribution. Unsupervised domain transfer such as Cycle GAN (Zhu et al., 2017; Kim et al., 2017) associate one domain $x$ distribution with another domain $y$ distribution.

In this paper, we consider the case of associating two distributions $q(x)$ and $q(z)$. Let $p(x|\theta)$ and $p(z|\phi)$ are learning models which have parameter $\theta, \phi$. Then, the Kullback-Leibler divergence from the target distribution to the learning model is given by

$$KL^x(\theta) = \int q(x) \log \frac{q(x)}{p(x|\theta)} dx, \ KL^z(\phi) = \int q(z) \log \frac{q(z)}{p(z|\phi)} dz.$$

In order to associate the two distributions, we restrict the learning models by the following equations,

$$p(x|\theta) = \int p(x|z,\theta)q(z)dz, \ p(z|\phi) = \int p(z|x,\phi)q(x)dx,$$

where $p(x|z,\theta), p(z|x,\phi)$ are conditional distributions (called generator and encoder). Then, using variational principle, the above KL divergence is upper bounded by the KL divergence of the joint distributions of $(x, z)$.

$$KL^x(\theta) = \int q(x) \log \frac{q(x)}{\int p(x|z,\theta)q(z)dz} dx = \int q(x) \log \frac{q(x)}{\int p(x|z,\theta)q(z)\frac{p(z|x,\phi)}{p(z|x,\phi)}dz} dx$$

$$\leq \int \int p(z|x,\phi)q(x) \log \frac{p(z|x,\phi)q(x)}{p(x|z,\theta)q(z)} dzdx = KL(p(z|x,\phi)q(x)||p(x|z,\theta)q(z))$$

$$KL^z(\phi) = \int q(z) \log \frac{q(z)}{\int p(z|x,\phi)q(x)dx} dz = \int q(z) \log \frac{q(z)}{\int p(z|x,\phi)q(x)\frac{p(x|z,\theta)}{p(x|z,\theta)}dx} dz$$

$$\leq \int \int p(x|z,\theta)q(z) \log \frac{p(x|z,\theta)q(z)}{p(z|x,\phi)q(x)} dxdz = KL(p(x|z,\theta)q(z)||p(z|x,\phi)q(x))$$

where we use Jensen's inequality. We define the objective function as the sum of the upper bounds.

$$KL^x(\theta) + KL^z(\phi) \leq KL(p(z|x,\phi)q(x)||p(x|z,\theta)q(z)) + KL(p(x|z,\theta)q(z)||p(z|x,\phi)q(x))$$

which is symmetric KL derivation of ALI (Dumoulin et al., 2016; Pu et al., 2017).

## 2  MARGINAL AND CONDITIONAL MATCHING

As described by previous section, ALI matches the two implicit joint distributions $p(z|x, \phi)q(x)$ and $p(x|z, \theta)q(z)$. If the two joint distributions match exactly, their marginal and conditional distributions must also match. However, ALI does not have the terms which directly match marginal and conditional distributions. Therefore, ALI has limitation on the reconstruction of real samples and cause the mode missing behavior (Li et al., 2017; Belghazi et al., 2018). For this problem, we utilize the following equations.

First, for marginal distribution, we integrate out the $p(z|x, \phi)q(x)$ and $p(x|z, \theta)q(z)$ by $x$ and $z$,

$$\int p(z|x, \phi)q(x)dx = p(z|\phi), \ \int p(x|z, \theta)q(z)dx = q(z),$$

$$\int p(x|z, \theta)q(z)dz = p(x|\theta), \ \int p(z|x, \theta)q(x)dz = q(x).$$

These marginal matchings are similar to GAN's objectives (Difference is not JS but reverse KL.).

Second, for conditional distribution, we obtain the following result by Bayes' theorem,

$$\log \frac{p(z|x, \phi)}{p(z|x, \theta)} = \log \frac{p(z|x, \phi)}{\frac{p(x|z,\theta)q(z)}{p(x|\theta)}} = \log \frac{p(z|x, \phi)p(x|\theta)}{p(x|z, \theta)q(z)}. \tag{1}$$

$$\log \frac{p(x|z, \theta)}{p(x|z, \phi)} = \log \frac{p(x|z, \theta)}{\frac{p(z|x,\phi)q(x)}{p(z|\phi)}} = \log \frac{p(x|z, \theta)p(z|\phi)}{p(z|x, \phi)q(x)}, \tag{2}$$

For example, in equation (2), for matching conditional distributions, we should match the joint distribution $p(x|z, \theta)p(z|\phi)$ and $p(z|x, \phi)q(x)$. In other words, we should match the pair of (reconstruction $x$, embedding $z$) and (real $x$, embedding $z$).

Using these observations, we propose the new objective of the matching which adds marginal and conditional distribution matching to ALI (Dumoulin et al., 2016). It is noted that our proposal is completely implicit and our conditional matching is different from Alice (Li et al., 2017) which discriminates the pair (real x, real x) and (real x, reconstruction x).

$$\begin{aligned} Match(\theta, \phi) \equiv &KL(p(z|x, \phi)q(x)||p(x|z, \theta)q(z)) + KL(p(x|z, \theta)q(z)||p(z|x, \phi)q(x)) \\ &+ KL(p(z|\phi)||q(z)) + KL(p(x|\theta)||q(x)) \\ &+ KL(p(z|x, \phi)p(x|\theta)||p(x|z, \theta)q(z)) + KL(p(x|z, \theta)q(z)||p(z|x, \phi)p(x|\theta)) \\ &+ KL(p(x|z, \theta)p(z|\phi)||p(z|x, \phi)q(x)) + KL(p(z|x, \phi)q(x)||p(x|z, \theta)p(z|\phi)) \end{aligned}$$

## 3  RATIO ESTIMATOR

In the case of implicit distributions, although we can obtain samples from the distributions, we cannot directly evaluate a term such as $KL(p(x|\theta)||q(x))$. For this problem, we use the probabilistic classification method of the ratio estimation (Sugiyama et al., 2012; Uehara et al., 2016; Mohamed & Lakshminarayanan, 2016; Tran et al., 2017). We introduce label $y$ and let $p(x|\theta) \equiv p(x|y =' numer')$ and $q(x) \equiv p(x|y =' denom')$. Using Bayes theorem,

$$r(x) \equiv \frac{p(x|\theta)}{q(x)} = \frac{p(x|y =' numer')}{p(x|y =' denom')} = \frac{p(y =' denom')p(y =' numer'|x)}{p(y =' numer')p(y =' denom'|x)},$$

$$= \frac{n_{'denom'}p(y =' numer'|x)}{n_{'numer'}p(y =' denom'|x)},$$

is derived, where $n_{'denom'}$ and $n_{'numer'}$ are the number of samples from each distributions. We choose $p('numer'|x) \equiv \sigma(c(x))$ where $c(x) \in \mathbf{R}$ is deep neural networks that have scalar output and $\sigma(x)$ is sigmoid function $\frac{1}{1+\exp(-x)}$. Then,

$$r(x) = \frac{n_{'denom'}\sigma(c(x))}{n_{'numer'}(1 - \sigma(c(x)))}.$$

Especially, when the case of $n_{'numer'} = n_{'denom'}$, we can estimate

$$\log r(x) = \log \frac{\frac{1}{1+\exp(-c(x))}}{1 - \frac{1}{1+\exp(-c(x))}} = c(x).$$

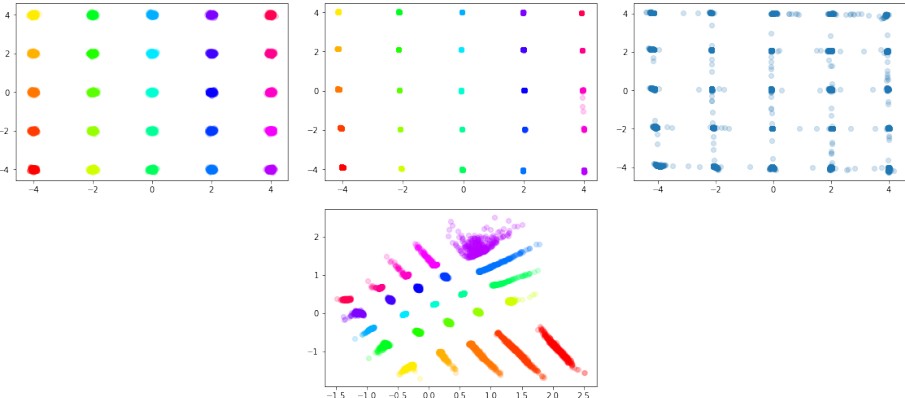

Figure 1: From the top left, real data, reconstruction, random generation, embedding.

Table 1: Inception Score and FID in CIFAR-10

| method | Inception Score | FID |
|---|---|---|
| sn-dcgan | 7.5 | 23.6 |
| proposed | 7.5 | 21.8 |

## 4 EXPERIMENTS

We conducted three experiments. In all experiments we modified the code of chainer-gan-lib[1] and used their spectral normalization discriminator (Miyato et al., 2018) as our ratio estimator.

First, by simple Gaussian mixture data, we compared the random generation and reconstruction quality. Figure 1 shows that proposed method simultaneously achieves reconstruction and random generation which are comparable to MINE and better than ALI and ALICE (See Figure 5 in Belghazi et al. (2018)).

Second, by CIFAR-10 (Krizhevsky et al., 2012) data, we compared the inception score (Salimans et al., 2016) and FID (Heusel et al., 2017) with chainer-gan-lib's sn-dcgan which has the highest score. Table 1 shows that the additional conditional terms do not degrade the quality of random generation.

Third, by CelebA (Liu et al., 2015) data, Figure 2 shows proposed method achieve reasonable reconstruction and random generation in the fully implicit manner. It is noted that the errors of proposed method do not stick to the pixels of the image and the output images are not blurry as VAE.

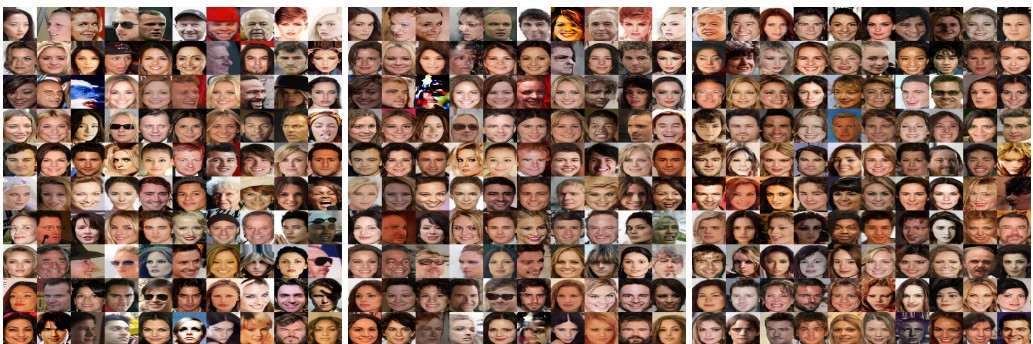

Figure 2: From the left, real data, reconstruction, random generation.

---

[1]https://github.com/pfnet-research/chainer-gan-lib

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
