# OpenReview forum: "Variational Implicit Distribution Matching"
_ICLR.cc/2018/Workshop — Reject_

### Official Review · AnonReviewer3 · 2018-03-03
**Contain some interesting ideas, but the presentation is a bit unclear**

**Rating:** 4
**Confidence:** 4

**Review:**

Thank you for an interesting read.

The paper improves on the adversarially learned inference (ALI) framework, which matches the joint distribution of x, z between the generative model and the encoding model. Instead of using KL, the authors propose using symmetric KL. Also to further improve performance the authors also suggest matching the marginal distributions and conditional distributions.

I think the idea of matching marginals, conditionals and joint distributions together can be useful. However the paper is not very clearly written, and also experimental study is somewhat weak.

1. How many discriminators you used for estimating the density ratios? My understanding is that you need 4?
2. What is sn-dcgan in Table 1?
3. What do you mean by showing the embedding in Figure 1?
4. How does your method compare to ALI/ALICE quantitatively?

I understand it's a bit strict to ask a workshop abstract to perform thorough experiments. But given the fact that we have so many variants of GANs (and many of them generates very good looking images), I think it's important to demonstrate the advantage of a new approach via clear quantitative analysis.

In summary, the abstract contains novel ideas, but in all doesn't quite meet the "late-breaking developments, very novel ideas and position papers" criteria.

---

### Official Review · AnonReviewer1 · 2018-03-12
**An extension of Symmetry-KL method by including marginal and conditional distribution matching**

**Rating:** 5
**Confidence:** 4

**Review:**

This paper proposes an extension of the Symmetry-KL method of Pu et. al.2017 by imposing marginal and conditional distribution matching within the framework.

One concern I have is that since the paper aggresses that if the two joint distributions match exactly, their marginal and conditional distributions must also match, why bother to introduce the marginal and conditional distribution matching components?

Also, I wonder how the marginal distribution matching above eq.1 is implemented? For example, to generate marginal samples x, from the equations, it seems it is done like this way: z ~ q(z), x ~ p(x|z,\theta) and x ~ q(x), z ~ p(z|x,\theta). Isn't this the way to draw samples for joint distribution matching?

For the ration estimator, not sure what 'numer' and 'denom' mean.

---

### Official Review · AnonReviewer2 · 2018-03-17
**Nice derivation for distribution matching; not clear what application is**

**Rating:** 5
**Confidence:** 4

**Review:**

The authors derive a symmetric KL derivation for ALI, which results in matching the joint distribution between p(x | z)q(z) and p(z | x) q(x), where (if I understand correctly) p(x | z) is an implicit model, q(z) is the prior, q(z | x) is the inverse of the generator, and q(x) is the empirical data distribution. They then argue that joint-distribution-matching does not lead to good reconstructions and causes missing modes; so they add to the objective function distribution matching to enforce the conditional and marginals.

The paper is not well-written; aside from the verbose mathematical derivations, it was hard to ground each of these distributions into something concrete in terms of a probability model, variational approximation, and data distribution. I recommend the authors make connections clearer to individual distributions.

I also am not convinced that joint-distribution-matching is why ALI has poor reconstruction and missing modes: please back this up with empirical evidence (or related literature).

Overall, the objective function seems overly complicated: Can you clarify the computional considerations, for example, how many neural networks/parameters, how many forward passes are required per neural network? Can you do an ablation study dropping individual terms of the objective to verify your claims about why conditionals or marginals are as important to include in the objective? Also, have you tried tuning hyperparameters which weight each of the terms to contribute to the objective?

---

### Decision · Program_Chairs · 2018-03-20
**ICLR 2018 Workshop Acceptance Decision**

**Decision:**

Reject

**Comment:**

Based on the reviews, this paper has not been accepted for presentation at the ICLR workshop. However, the conversation and updates can continue to appear here on OpenReview.